# Dietary (Poly)phenols in Traumatic Brain Injury

**DOI:** 10.3390/ijms24108908

**Published:** 2023-05-17

**Authors:** Rafael Carecho, Diogo Carregosa, Bernardo Oliveira Ratilal, Inês Figueira, Maria Angeles Ávila-Gálvez, Cláudia Nunes dos Santos, Natasa Loncarevic-Vasiljkovic

**Affiliations:** 1iNOVA4Health, NOVA Medical School, Faculdade de Ciências Médicas, NMS, FCM, Universidade Nova de Lisboa, 1169-056 Lisboa, Portugal; rafael.carecho@nms.unl.pt (R.C.);; 2ITQB, Instituto de Tecnologia Química e Biológica António Xavier, Universidade Nova de Lisboa, 2780-157 Oeiras, Portugal; 3Hospital CUF Descobertas, CUF Academic Center, 1998-018 Lisboa, Portugal; 4Clínica Universitária de Neurocirurgia, Faculdade de Medicina da Universidade de Lisboa, 1649-028 Lisboa, Portugal; 5iBET, Instituto de Biologia Experimental e Tecnológica, 2781-901 Oeiras, Portugal; 6Laboratory of Food & Health, Group of Quality, Safety, and Bioactivity of Plant Foods, CEBAS-CSIC, 30100 Murcia, Spain

**Keywords:** secondary brain injury, inflammation, oxidative stress, neurodegeneration

## Abstract

Traumatic brain injury (TBI) remains one of the leading causes of death and disability in young adults worldwide. Despite growing evidence and advances in our knowledge regarding the multifaceted pathophysiology of TBI, the underlying mechanisms, though, are still to be fully elucidated. Whereas initial brain insult involves acute and irreversible primary damage to the brain, the processes of subsequent secondary brain injury progress gradually over months to years, providing a window of opportunity for therapeutic interventions. To date, extensive research has been focused on the identification of druggable targets involved in these processes. Despite several decades of successful pre-clinical studies and very promising results, when transferred to clinics, these drugs showed, at best, modest beneficial effects, but more often, an absence of effects or even very harsh side effects in TBI patients. This reality has highlighted the need for novel approaches that will be able to respond to the complexity of the TBI and tackle TBI pathological processes on multiple levels. Recent evidence strongly indicates that nutritional interventions may provide a unique opportunity to enhance the repair processes after TBI. Dietary (poly)phenols, a big class of compounds abundantly found in fruits and vegetables, have emerged in the past few years as promising agents to be used in TBI settings due to their proven pleiotropic effects. Here, we give an overview of the pathophysiology of TBI and the underlying molecular mechanisms, followed by a state-of-the-art summary of the studies that have evaluated the efficacy of (poly)phenols administration to decrease TBI-associated damage in various animal TBI models and in a limited number of clinical trials. The current limitations on our knowledge concerning (poly)phenol effects in TBI in the pre-clinical studies are also discussed.

## 1. Introduction

Traumatic brain injury (TBI), a form of acquired brain injury, occurs when a sudden trauma causes damage to the brain. Although it has been historically ignored as a major health issue, TBI represents the leading cause of death and disability in young adults worldwide [1]. Globally, the annual incidence of TBI is variably estimated at 27 to 69 million [2] and it is predicted that about half of the world’s population will have one or more TBIs over their lifetime. World Health Organization (WHO) data indicate that the world is facing a ‘silent epidemic’ of TBI. Epidemiological predictions until 2030 are anticipating a two to three times higher incidence of patients suffering from TBI-induced disabilities than those with neurological ones, arising from Alzheimer’s disease, Parkinson’s disease, or cerebrovascular disorders [1]. Despite recent advances in TBI research and the current efforts of collaborative studies using multidisciplinary approaches to tackle this problem and improve patients’ outcomes, there is still no therapeutic solution for TBI, and it continues to represent a major global health burden and public health challenge among all ages worldwide.

### Different Types of TBI

TBI is defined as an alteration in brain function, or other evidence of brain pathology, caused by an external force [3]. It has been shown that TBI causes the imbalance of a multitude of metabolic, molecular, and biochemical functions, profoundly affecting brain homeostasis, leading to temporary or permanent neurocognitive deficits, motor disabilities, or even psychological disturbances [4,5]. The extent to which these mechanisms are disrupted in TBI depends on the severity of the impact on the brain. TBI severity is normally assessed by the Glasgow Coma Scale (GCS) [6,7,8,9] (Figure 1). TBI is classified accordingly with the GCS into mild (13–15), moderate (9–12), and severe (≤8) [7,9]. In addition to the GCS, TBI is clinically evaluated by several imaging techniques to determine the severity of structural damage to the brain tissue. In addition to the severity, TBI can be classified into closed brain injury or penetrating brain injury [10]. Closed brain injuries (concussions, contusions, brain hemorrhages, intracranial hematomas, diffuse axonal injury, and coup-contrecoup brain injury) are brain injuries with no disruption (fracture) of the skull, caused by a rapid forward or backward movement and shaking of the brain that results in bruising and tearing of the brain tissue [10,11,12]. Closed brain injuries are usually caused by car accidents, falls, and in contact sports. On the other hand, penetrating, or open head injuries, imply a disrupted (fractured) skull that in most cases involves injuries caused by a bullet or some other projectile [11].

When presented in numbers, only 20% of total patients suffer from moderate and severe TBI (approximately 10% for each type of TBI). The great majority of TBI patients (about 80%) suffer from mild injuries [13], which are linked to the fewer mortality rates as compared to both moderate and severe injuries [14]. Of the total number of mild TBIs, 20% are concussions, and 80% of all concussions are sports-related concussions [5] (Figure 1). According to the available epidemiological data, men are more prone to sustaining TBI, which is caused, in more than 50% of cases, by falls (most of them sport-related) and road traffic accidents [13,15,16].

## 2. Primary vs. Secondary Brain Injury

TBI is often mistakenly seen as an event. In reality, TBI is rather a process, as it consists of two distinct phases—primary and secondary brain injury—and it lasts far longer than the moment of physical impact itself. Primary injury refers to the sudden and profound injury to the brain, causing mechanical harm to the tissue at the moment of the impact [17,18]. Although the primary insult occurs in a short time span of approximately 100 milliseconds, this initial insult triggers secondary brain injury. Secondary brain injury refers to the repertoire of changes that evolve over time—from hours and days to weeks and months, and sometimes even years after the insult—leading to biochemical, cellular, and physiological changes, such as the further disruption of the blood–brain barrier (BBB), neuroinflammation, excitotoxicity, axonal damage, necrosis, apoptosis, mitochondrial dysfunction, and generation of oxidative stress [18,19,20,21,22] (Figure 2). All these processes are intertwined, and each of them is contributing to the further destruction of affected brain tissue, adding to the secondary brain injury progression and poorer outcome after the TBI. 

Importantly, neuronal damage after TBI appears in two phases, primary cell death, induced directly by the initial impact, and delayed secondary death of neurons, the latter depending on environmental changes, lack of metabolic and trophic supply, and altered gene transcription [23]. A multitude of diverse secondary cascades detrimental to axons have been investigated following TBI. Specifically, alterations to mitochondria, oxidative stress, and lipid peroxidation have long been implicated in cytoskeletal degradation in vivo [24,25,26,27,28,29].

Increasing evidence suggests that neuroinflammation and microglial activation in the white matter may also contribute to cellular damage [30] and, remarkably, can persist for years after injury in humans [29,31,32]. All the processes of secondary brain injury are, to some extent, implicated in secondary neuronal and glial cell death. Neuronal cell death after TBI is a major cause of neurological deficits and mortality [33,34]. While the primary death of neurons occurs within a short time after trauma, secondary cell death, as a consequence of secondary brain injury, lasts for weeks, months, and even years after the TBI. In terms of TBI patient management, currently, there are no procedures and therapies to reverse the initial insult causing a TBI [35]. Therefore, management strategies are focused on preventing and/or alleviating secondary injury. However, one should keep in mind that since a variety of processes occur after the TBI, any treatment developed to prevent or mitigate secondary brain injury will need to tackle more than one of these processes to be considered successful.

### Current Pharmacological Approaches for Treating TBI Patients

In clinical settings, TBI still represents a great challenge for treatment since it is a complex condition. Treatment for TBI may involve medication therapy and rehabilitation, depending on the extent of the injury. To minimize cerebral injury after the TBI, therapeutic intervention is directed to attenuate the first impact damage and to restrict the molecular and cellular cascade of continuous cell damage. So far, there are no effective treatments for the first impact damage [36]. Therefore, many studies have been carried out to search for treatment to prevent further (secondary) neuronal damage after the TBI and to enhance neural network reorganization and functional recovery. Unfortunately, these experimental studies have not been successfully translated into clinical therapies.

Numerous pharmacological therapies are in use for the treatment of TBI patients. For more than 30 years, corticosteroid drugs have been widely used to treat patients with a brain injury [37]. However, because of a significant increase in the mortality rate in patients given steroids compared with patients who received no treatment [38,39], numerous alternative pharmaceuticals arouse as better and safer options for the treatment of TBI patients, such as progesterone, erythropoietin, tranexamic acid, mannitol, nimodipine, ziconotide, ramelteon, and citicoline [40].

Despite several decades of successful pre-clinical studies and very promising results for the above-mentioned therapies, when transferred to clinics they showed, at best, a modest beneficial effect, but more often, an absence of effects or even very harsh side effects, when administered to TBI patients [40,41]. This universal failure has highlighted the need for novel approaches that will be able to respond to the complexity of the TBI and tackle TBI pathological processes on multiple levels. Recent evidence strongly indicates that nutritional support following the TBI in terms of early initiation of nutrition, taking special care of the optimal glucose, protein, and lipid concentrations needed for neuronal survival, may provide a unique opportunity to ameliorate secondary injury and improve outcomes after the TBI [42,43,44,45]. More and more pre-clinical studies appear to show that simple, non-invasive changes in the everyday diet may significantly affect secondary brain injury on multiple levels, leading to a better recovery after the TBI [4,5,46,47,48,49].

## 3. Dietary (Poly)phenols in TBI Prevention and Management

Although nutritional approaches encompass a broad range of molecules such as minerals and vitamins, dietary (poly)phenols, a big class of molecules, have emerged in the past few years as promising compounds for TBI due to their range of pleiotropic effects. The term (poly)phenols represents a vast group of chemical molecules, classically of natural origin, e.g., flavonoids, ellagitannins and ellagic acid, coumarins, stilbenes, phenolic acids, and many others (Figure 3). (Poly)phenols are present in large quantities in fruits and vegetables, tea, wine, spices, cocoa, and coffee. Since (poly)phenols are almost ubiquitous in plant foods and daily beverages (e.g., tea and coffee), plant-based diets or specific dietary patterns (e.g., Mediterranean diet) provide a significant amount of these bioactive compounds [50,51]. Upon ingestion, dietary (poly)phenols are degraded, starting in the stomach mainly with the loss of their glycoside group. Their absorption continues throughout the small and large intestines [52]. In the gut, the microbiota can further metabolize and break some (poly)phenols into smaller entities, leading to the generation of a vast array of lower molecular weight (poly)phenol metabolites [53,54]. Altogether, some (poly)phenols can cross the epithelial barrier and reach the liver, undergoing phase II conversion such as sulfate, glucuronide, methyl, and glycine conjugation before entering systemic circulation [52,53,54]. Noteworthy, (poly)phenol metabolites are those that can be found in circulation at considerably higher concentrations than their parent counterparts, presenting a higher capacity to be absorbed and reach the target tissues, including the brain [55,56,57,58]. By doing so, (poly)phenol metabolites may be the true effectors for many of the systemic effects of (poly)phenol-rich foods.

It is important to mention that, besides the systemic effects that (poly)phenols and their bioavailable metabolites arising from our diet may hold even if they are not BBB permeable, for a better understanding of their role in the prevention and/or treatment of brain disorders, BBB permeability should be understood. Indeed, we and others have evidenced the BBB transport of (poly)phenols and of bioavailable circulating metabolites at physiologically relevant concentrations [56,59,60,61], both in vitro and in vivo, as reviewed elsewhere [55,62]. Such data clearly reinforce dietary (poly)phenols’ potential to reach brain cells, and, ultimately, may help to explain the molecular mechanisms behind their role as putative brain health-promoting compounds in a nutritional context.

(Poly)phenol metabolites have been shown as very potent regulators of neuroinflammation, oxidative stress, and neuronal cell death in neuronal and microglial in vitro models [53,56,63,64,65,66]. 

(Poly)phenols and their metabolites have also been described as effective in increasing cell viability, by decreasing reactive oxygen species (ROS) levels and positively modulating the balance of anti-/pro-apoptotic proteins and glutathione metabolism both in in vitro (SH-SY5Y cells and 3D cell model of Parkinson’s disease) and in vivo models of neurodegenerative diseases [67,68]. Regarding in vivo studies, (poly)phenols also revealed very promising results in counteracting inflammation, oxidative stress, and neurodegeneration in ischemic animal models, decreasing both microgliosis and astrogliosis [69,70]. In addition, at the functional level, significant memory and cognitive improvement in the animal models for neurodegeneration [71] and aging [72] were observed when animals were treated with (poly)phenol compounds. In a 1-methyl-4-phenyl-1,2,3,6-tetrahydropyridine (MPTP) mouse model of Parkinson’s disease, the improvement in motor functions was observed upon treatment with the syringic acid [73].

Among dozens of experimental designs, the increasing evidence regarding the neuroprotective and multitargeted potential of (poly)phenols to alleviate or even prevent both neuroinflammation and oxidative stress processes is indubitable. Moreover, (poly)phenols metabolites appear in circulation, being able to cross the BBB and reach the brain while reporting no major side effects, which makes them very appealing as potential bioactive compounds in both preventive and therapeutic settings. Currently, the focus on (poly)phenols as a potential therapeutic application in TBI is growing due to their safety and pleiotropic mechanism of action.

### 3.1. (Poly)phenols in TBI State-of-the-Art from the In Vivo Models

Several TBI rodent models are routinely used in pre-clinical settings, having different clinical relevance. Controlled cortical impact (CCI) and fluid percussion injury (FPI) are models mimicking sport-related TBI in humans by inducing cerebral contusion and concussion [74,75]. The weight-drop model is the most used model, since it mimics the mechanisms of TBI in humans caused by falls and car accidents. On the other hand, the blast wave model is similar to the injuries that occur in soldiers in the battlefield [74,75].

So far, (poly)phenols used in TBI studies undoubtedly showed beneficial effects on the main processes underlying secondary injury following the TBI. Most of the studies in rodents, regardless of the TBI model applied, used the intraperitoneal (IP) administration route as a route of choice to deliver compounds, while fewer studies used an oral gavage or nutritional/food approach (Figure 4 and detailed information in Appendix A). Regardless of the route of administration, the dose range of (poly)phenols used in the studies is quite wide (0.1–300 mg/kg), while the most used doses were 50 and 100 mg/kg (referring to an 8 and 16 mg/kg, respectively, when transferred to a human dose) [76,77].

Moreover, there is also a difference in the application regimen, since in most of the studies, the administration of compounds started right after the TBI. Only a very few studies used the preventive approach, administering compounds before the TBI. In this pre-treatment paradigm, the administration was performed at various time points, ranging from 15 min to 4 weeks before the brain injury [78,79,80,81,82,83,84,85,86,87,88,89]. In addition, very few studies comparing pre-, post-, and continuous treatment with (poly)phenols were performed, and these are essential to acquire the information about the most effective paradigm.

What also should be considered is that different routes and different doses represent different approaches. In that sense, intraperitoneal, intravenous, intranasal, or subcutaneous routes will always represent a pharmacological approach. This is because if (poly)phenols are applied by any of the mentioned routes, they are skipping the phase I metabolism in the intestine, meaning that all the (poly)phenol metabolites that would emerge during this process are lost. Since these metabolites are only appearing in circulation upon the (poly)phenol ingestion and their metabolism by gut microbiota, the routes that can be considered as nutritional approaches are through the oral gavage, food, or water. Only if food-achievable doses of (poly)phenols are applied can one consider it as a nutritional approach. However, we may also consider a nutraceutical approach when through supplements, the doses normally exceed the amounts that could be ingested by normal daily food intake. Regardless of the dose, route, and/or the application paradigm, all studies undoubtedly showed similar beneficial effects of (poly)phenols on major processes of secondary injury.

#### 3.1.1. Flavanols

Catechin, (-)-epicatechin (EC), and (-)-epigallocatechin-3-gallate (EGCG) were the only flavanols studied in TBI models presenting pleiotropic effects affecting different pathological aspects of TBI (Figure 4 and Appendix A). Both catechin and EC treatment following TBI significantly reduced brain edema formation [90,91] as well as brain tissue damage [90,91]. Both EC and EGCG have shown significant neuroprotective effects with a reduction in the apoptotic cell death of the neurons surrounding the injured area [78,79,80,91]. Moreover, EGCG has been shown to induce an increase in the number of neural stem cells (nestin-positive cells) in the peri-lesion area [79]. Although the study by Itoh and colleagues (2013) demonstrated that treatment with EGCG significantly increases the number of B-cell lymphoma 2 (Bcl2)-positive neuronal cells around the injured area [80], showing a direct anti-apoptotic effect of this flavanol, it should be taken into consideration that the effects on the other processes of secondary brain injury can also indirectly, but substantially, affect neuronal cell death. In that sense, all three flavanols exploited in the TBI paradigm were shown to be potent modulators of neuroinflammation and oxidative stress. Catechin treatment suppressed the local inflammatory response in the ipsilateral brain after TBI, significantly reducing the expression of the mRNA of pro-inflammatory factors such as IL-1β, inducible nitric oxide synthase (iNOS), and IL-6, whereas it increased the mRNA expression of the anti-inflammation-associated factor (arginase 1) [90]. Moreover, EC treatment substantially reduced neutrophil infiltration into the injured tissue after the TBI [91], while EGCG treatment showed reduced ionized calcium-binding adapter molecule 1 (Iba-1) and glial fibrillary acidic protein (GFAP) mRNA expression and reduced number of activated microglial and astrocytic cells in the peri-injured area, as well as downregulated tumor necrosis factor-alpha (TNF-α), IL-1β, IL-6 serum, and brain protein levels in the peri-injured area [92]. The potential mechanism of the anti-inflammatory action of EGCG, as revealed by Wu and Cui (2020), is a reduction in the phosphorylation of the nuclear factor of the kappa light polypeptide gene enhancer in the B-cells inhibitor, alpha (IκBα), and IκB kinase α (IKKα) and IKKβ, and decreased nuclear translocation of nuclear factor kappa B (NF-κB) p65 subunit [92].

In terms of oxidative stress and oxidative damage, 4 weeks of EGCG pre-treatment has been shown to significantly decrease malondialdehyde (MDA) levels as well as alleviate lipid peroxidation and oxidative damage, 1 and 3 days after the TBI [79]. In addition, post-treatment with EGCG has been shown to be equally protective by reducing O2-, H2O2, and MDA, and increasing antioxidant enzyme—SOD, CAT, and glutathione (GSH)—levels in the hippocampus (Hpp) 7 days after the TBI [92]. Moreover, EC has also been shown to alleviate reactive oxygen species (ROS) production and decrease matrix metallopeptidase 9 (MMP-9) enzyme activity, reduce Kelch-like ECH-associated protein 1 (Keap 1) expression while increasing nuclear factor-erythroid factor 2-related factor 2 (Nrf2) nuclear accumulation. Subsequently, EC increases SOD and NAD(P)H:quinone oxidoreductase 1 (NQO1) expression and reduces heme oxygenase-1 (HO-1) expression and iron deposition without reducing the brain hemoglobin level 3 days after the TBI [91,92]. The study by Wu and Cui (2020) revealed that EGCG regulates inflammation and oxidative stress through the 5’ adenosine monophosphate-activated protein kinase (AMPK) pathway since all the EGCG effects were completely diminished in AMPKα1-knockout mice [92]. The study by Cheng and colleagues showed that EC protects the TBI brain by activating the Nrf2 pathway, inhibiting HO-1 protein expression, and reducing iron deposition. The latter two effects could represent an Nrf2-independent mechanism in this model of TBI [91], meaning that protection by EC involves Nrf2-dependent and -independent pathways.

Moreover, catechin treatment after the TBI in rats preserved BBB integrity 24 h after the TBI by the alleviation of the TBI-induced loss of tight junction proteins as revealed by the decreased mRNA expression and protein levels of zonula occludens 1 (ZO-1) and occludin in the ipsilateral cortex [90].

Of utmost importance is that all three flavanols tested in the TBI paradigm significantly improved long-term neurological outcomes after the TBI. Catechin treatment improved motor performance [90], EC improved the neurologic deficit score [91,92], and EGCG significantly improved neurological function after the TBI [92]. Regardless of the regimen of application (prevention or treatment of TBI), all studies on flavanols reviewed herein showed undoubtedly that these compounds can effectively alleviate all major processes of the secondary injury (neuroinflammation, oxidative stress, neurodegeneration), leading to the improved functional outcome after the TBI. In this sense, evidence suggests that both a preventive lifestyle prior to injury, and introduction of (poly)phenols after the TBI as a therapeutic treatment, will result in an improved outcome. However, one comparative study, using pre-, continuous, and post-TBI treatment, showed that EGCG significantly decreased MDA levels 1 and 3 days after the TBI with the most significant decrease in the continuous treatment group [80]. Moreover, it also described that the greatest reduction in apoptosis, DNA damage, oxidative damage, and lipid peroxidation in the continuous treatment occurred on days 1, 3, and 7 after the TBI [80]. Although in all EGCG treatment paradigms anti-apoptotic protein Bcl2 was elevated on days 1 and 3 after the TBI, the highest increase was detected in the continuous treatment group [80]. Glial scar formation was also reduced on day 7 in the continuous and post-treatment groups [80]. Significant improvement in cognitive impairment after the TBI was only observed in the continuous and post-TBI in the EGCG treatment groups [80]. These data highlight the importance of the treatment regimens of (poly)phenols, indicating that continuous oral application is the regimen that consistently gives the most pronounced neuroprotective, anti-inflammatory, anti-oxidative stress, and phenotypic effects. 

In all the studies that tested the effects of flavanols in the TBI models described herein, compounds were given orally and in doses that are nutritionally relevant. None of these studies, however, performed metabolomic analysis to determine what flavanols (or their metabolites) and in which concentrations reached the brain. Although we may consider some indirect systemic effects, it is important to describe the neuroprotective effects for the (poly)phenol metabolites [53,55].

#### 3.1.2. Flavanones

Hesperidin and naringenin were the only flavanones tested in the TBI paradigm and treatment was exploited although in different experimental approaches (Figure 4 and Appendix A). Hesperidin was tested in a nutritional approach and therefore it is expected that what reaches circulation and consequently the brain could be its metabolites. In fact, in plasma, concentrations of hesperidin aglycone, and hesperidin were predicted to be <0.02 mg/L at an oral dose of 50 mg/kg of hesperidin by body weight in humans [93]. On the other hand, the naringenin trial considered pharmacological administration by IP, avoiding the gut metabolism of this compound.

Hesperidin treatment, by oral administration, has been shown to significantly reduce depression-related symptoms in mice subjected to mild TBI. Since hesperidin decreased the levels of IL-1β, and MDA, and increased brain-derived neurotrophic factor (BDNF) levels in the hippocampus, the study by Kosari-Nasab and colleagues suggests that the observed anti-depressant-like effect of hesperidin may be mediated, at least in part, by decreased neuroinflammation and oxidative damage, and enhanced BDNF production in the hippocampus [94].

Naringenin treatment by intraperitoneal injection has also been shown to be effective in alleviating brain edema in both tested doses (50 and 100 mg/kg) with the dose of 100 mg/kg showing better effects. Naringenin (100 mg/kg) improved the neurological severity score (mNSS) at 3, 5, and 7 days post-TBI, while the dose of 50 mg/kg induced the improvement in mNSS only 3 days post-TBI [94]. At the molecular level, naringenin, in a dose of 100 mg/kg, reduced TBI-induced neuronal cell death and apoptosis by increasing anti-apoptotic (Bcl2) and decreasing pro-apoptotic (Bcl2 associated X (Bax) and caspase-3) proteins. Moreover, it alleviated oxidative stress by reducing MDA generation and upregulation of glutathione peroxidase (GPx) activity, but also by reducing the expression of proteins associated with endoplasmic reticulum (ER) stress (Activating Transcription Factor 4 (ATF4), Glucose-regulated protein (GRP78), and CHOP) while slightly upregulating levels of phosphorylated eukaryotic translation initiation factor 2α (p-eIF2α) in the injured cortex of mice subjected to TBI [94].

#### 3.1.3. Flavonols

Four members of the flavonol family, quercetin, fisetin, rutin, and kaempferol, have been tested in the TBI setting so far (Figure 4 and Appendix A). Interestingly, both oral and IP administration was used for quercetin and its glycoside rutin (quercetin-3-O-rutinoside), allowing for some comparative comments concerning the effects of parent compounds versus their metabolites. Moreover, since rutin can only be deglycosylated to quercetin aglycone by enzymes from the gut microbiota, we may have expected a slower absorption of quercetin metabolites derived from rutin [95].

Quercetin has been shown to attenuate the inflammatory response after TBI by alleviating microgliosis, reducing TNF-α, iNOS, IL-1β, and IL-6 mRNA, and protein levels in the serum and/or brain [96,97,98,99], but also by increasing IL-10 levels [97]. Studies also showed a strong impact of quercetin, fisetin, and rutin on oxidative stress by restoring CAT, GPx, GSH, MDA, and SOD levels in the brain after the TBI [96,97,100,101,102,103] and decreasing lipid peroxidation [97,101] and myeloperoxidase (MPO) activity [101]. Moreover, restored cytochrome c in mitochondria, mitochondrial membrane potential, and intracellular ATP content together with the reduced mitochondrial lesions were reported upon quercetin treatment [100,104]. Since several studies observed increased Nrf2 levels and increased Nrf2 translocation to the nucleus in the brain upon the quercetin and fisetin treatment [96,100,103], it has been suggested that these flavonols exert their protective effects on TBI-induced inflammation and oxidative stress through activating the Nrf2 pathway and the increased expression of Nrf2 downstream proteins—HO-1 and NQO-1 [96,103]. Quercetin administration was also shown to stimulate mitochondrial biogenesis by the activation of the peroxisome proliferator-activated receptor-gamma coactivator-1α (PGC-1α) pathway [104].

Fisetin has been shown to protect the integrity of the BBB and reduce the brain lesion size [103], while quercetin, fisetin, and rutin have reduced neurodegeneration and apoptosis in the peri-lesioned region, as revealed by the increased number of living neurons, reduction in the active caspase-3 levels, and reduced number of apoptotic cells [97,102,103,104,105]. As a possible mechanism underlying the quercetin-induced neuroprotection, increased Akt phosphorylation, activation of the PI3K/Akt signaling pathway, as well as the decrease in ERK ½ phosphorylation [105,106], and reduction in Bax and induction of Bcl2 as main pro-apoptotic and anti-apoptotic proteins (respectively) have been suggested [97,102,104,106]. Moreover, neuronal autophagy in the hippocampus was reduced upon the quercetin treatment as was revealed by LC3/NeuN double labelling [106].

At the macroscopic level, quercetin, fisetin, and rutin successfully reduced brain edema as measured by the brain water content [96,98,102,103,104,105], and markedly improved neurological, motor, sensory–motor, and cognitive function that were substantially impaired following the TBI [97,99,102,103,105,106,107]. A study by Kosari-Nasab and colleagues has shown that quercetin decreases anxiety-related symptoms [108]. This finding was correlated with the quercetin-induced attenuation of the hypothalamic–pituitary–adrenal (HPA) axis hyperreactivity through the reduction in elevated serum levels of adrenocorticotropic hormone (ACTH) and corticosterone [109] as a potential mechanism underlying its anxiolytic action.

In the study by Parent and colleagues, kaempferol treatments were performed in the first 48 h post-TBI, and their effects were assessed by functional magnetic resonance imaging (fMRI) and diffusion tensor imaging (DTI) at adolescence (2 months post-injury). TBI prognosis was significantly altered at adolescence by early kaempferol treatment, with improved neural connectivity, neurovascular coupling, and parenchymal microstructure in select brain regions, with the highest changes in the frontal and parietal cortices and hippocampus [95].

All the TBI studies using flavonols presented here, regardless of the administration route, were considered post-TBI application, so no data are available about prevention or continuous treatment, in terms of a comparison of the effectiveness of these three different paradigms. Concerning the administration route, it is very interesting to note that independently of the route of compound delivery (oral versus IP), an improvement in TBI outcomes is described. This reflects that both parent flavonols and their metabolites could be contributing to the observed effects since that by oral gavage, the bioavailability of flavonols is very low. Moreover, the difference in the pharmacokinetics described for quercetin and its glycoside rutin is also a factor that does not reflect in the differences in the improvements in the TBI trials performed [95]. Additionally, the fact that different application regimens were used in the studies reinforces that independent of doses and time after the TBI, the tested flavonols are effective in the attenuation of TBI outcomes.

#### 3.1.4. Flavones

Luteolin, one of the most abundant dietary flavones, and chrysin were the only flavones exploited in TBI settings so far (Figure 4 and Appendix A). Absorption and metabolism studies pointed out the very low bioavailability of both flavones and this could be the reason for most of the studies performed for luteolin considering IP administration [110].

Luteolin, applied by IP both before and after the induction of TBI, was effective in alleviating brain inflammation as revealed by reduced TNF-α and IL-1β mRNA and protein levels, and reduced NF-kB p65 nuclear accumulation [81,111]. Luteolin restored MDA levels and GPx activity 24 h after the TBI [112]. A study by Xu and colleagues showed that luteolin enhanced the translocation of Nrf2 to the nucleus, subsequently upregulating the expression of the downstream factors such as HO1 and NQO1 [112]. Since luteolin treatment failed to provide neuroprotection after the TBI in Nrf2^-/-^ mice, these data suggest that luteolin exerts its neuroprotective effects in TBI settings, possibly through the activation of the Nrf2–antioxidant responsive element (ARE) pathway [112]. Luteolin has also been shown to reduce neuronal cell degeneration and apoptosis 24 h after the TBI, and to induce beclin1 and LC3 levels, indicating increased levels of autophagy [111,112]. Moreover, since TBI is recognized as a major risk factor for Alzheimer’s disease (AD), an interesting study by Sawmiller and colleagues performed on Tg2576 mice overexpressing Amyloid β (Aβ) protein as a model system for AD, showed that significant increases in Aβ deposition, glycogen synthase-3 (GSK-3) activation, and phospho-tau protein levels observed 3 days after TBI can be successfully ameliorated by 15 days of luteolin pre-treatment [81]. On the macroscopic level, luteolin significantly reduced the formation of brain edema 24 h after the TBI [111,112], and alleviated the neurological deficit induced by the TBI [112].

Chrysin treatment effects for TBI were only analyzed in one study after oral gavage and after the induction of the TBI. Like luteolin, it exerted strong effects on the level of oxidative stress after the TBI, significantly increasing SOD, CAT, GPx, and GSH levels, and significantly decreasing the MDA content 3 days after the TBI, and completely restored the levels of these proteins 14 days after the TBI in all the tested doses [113]. Chrysin showed dose-dependent effects on the reduction in the number of apoptotic cells and Bax protein, and the induction of the Bcl2 protein in the cerebral cortex and CA3 region of the hippocampus 14 days after the TBI. Moreover, chrysin improved learning and memory disabilities and ameliorated motor coordination impairment while no improvement in the Veterinary Coma Scale (VCS) was observed upon treatment by any dose of chrysin after the TBI [113].

The TBI studies using luteolin presented herein were applied either pre- or post-TBI treatment, with the IP route of administration. Both pre- and post-treatment showed obvious effects on the main processes of secondary brain injury [81,111,112]. To the best of our knowledge, unfortunately, no studies applying continuous treatment have been performed so far.

#### 3.1.5. Isoflavones

There has only been one study so far describing the effects of the isoflavone family members in the TBI model (Figure 4 and Appendix A). A study by Soltani and colleagues showed that genistein in a low dose (15 mg/kg) given after the TBI is capable of reducing brain edema, as determined by the significant reduction in the brain water content. Genistein also reduced intracranial pressure and increased the BBB integrity. Moreover, in terms of a functional outcome, genistein improved motor function [114].

#### 3.1.6. Curcuminoids

Curcumin as a representative (poly)phenol of the curcuminoid family has been extensively tested in TBI settings (Figure 4 and Appendix A). It has been associated with a significant attenuation in the levels of IL-1β, IL-6, IL-18, and TNF-α mRNA and protein levels, but also with a reduced microglial and astrocytic activation after the TBI [85,115,116]. The study by Zhu and colleagues showed that curcumin administration may improve TBI outcomes by reducing acute activation of microglia/macrophages and neuronal apoptosis through a mechanism involving the Toll-like receptor 4 (TLR4)/MyD88/NF-κB signaling pathway in the microglia/macrophages in TBI [117].

Moreover, curcumin decreased the levels of the oxidative stress marker—MDA, the principal product of polyunsaturated fatty acid peroxidation [84,116]. A study by Wu and colleagues showed that the supplementation of curcumin in the diet dramatically reduced oxidative damage and normalized levels of BDNF, synapsin I, and CREB that had been altered after the TBI [83]. Curcumin treatment also showed the potential to regulate molecules involved in energy homeostasis following the TBI by increasing the levels of AMP-activated protein kinase (AMPK), ubiquitous mitochondrial creatine kinase (uMtCK), and cytochrome c oxidase II (COX-II) [82].

Dong and colleagues showed that post-injury treatment with curcumin had neuroprotective effects by reducing the number of apoptotic neurons in the injured cortex, alleviating cleaved caspase-3 levels, and inducing Bcl2 protein expression [116]. These effects were aligned with the improved nuclear translocation of Nrf2 and upregulation of antioxidant enzymes in the Nrf2-ARE pathway, e.g., HO-1, NQO-1, glutamate-cysteine ligase catalytic subunit (GCLC), and glutamate-cysteine ligase modifier subunit (GCLM), suggesting that curcumin effects are, at least in part, accomplished through the Nrf2 regulated antioxidant response [116]. Moreover, a recent study by Sun and colleagues discovered that curcumin treatment not only suppresses neuronal cell death after TBI, but it also induces obvious neurogenesis in the hippocampus of treated animals [118], recovering the loss of synaptic proteins synaptophysin (SYN) and post-synaptic density protein 95 (PSD-95) in the hippocampus [119].

On a macroscopic level, curcumin applied 5 days before the TBI in doses of 50 and 100 mg/kg significantly decreased the animal death rate after the TBI in both examined concentrations. However, it reduced the lesion volume in a dose-dependent manner, with the dose of 100 mg/kg being more effective [84].

Curcumin administered post-injury significantly reduced brain edema following the TBI [116,117]. The study by Laird and colleagues revealed that curcumin reversed the induction of aquaporin-4 (AQP4), which is an astrocytic water channel implicated in the development of cellular edema following head trauma, indicating a possible mechanism of action by which curcumin decreases brain edema development after the TBI [85]. In terms of the functional outcome after the TBI, studies showed that curcumin can provide protection from motor, sensory, reflex, and cognitive impairment following the TBI [83,84,85,117,118]. 

Although the curcumin application differs among the studies, in terms of route (oral or IP) and regimen applied (single or multiple doses, before or after the TBI), the results undoubtedly showed its ability to reduce neuroinflammation, oxidative stress, neurodegeneration, and to improve the functional outcome after the TBI in all experimental paradigms. Two studies applied a continuous curcumin treatment (four weeks of pre- and one week of post-treatment), with curcumin being added to the animal food and showed strong positive effects of curcumin on the TBI outcome [82,83]. Just one study, however, made a comparison between pre- and post-treatment with curcumin, concluding that pre-treatment with curcumin was more efficient in reducing brain edema than post-treatment [85].

#### 3.1.7. Stilbenes

Although stilbenes are a big subfamily of (poly)phenolic compounds, to the best of our knowledge, resveratrol has been the only member used so far in TBI studies (Figure 4 and Appendix A). Strong evidence, however, undoubtedly shows that resveratrol has anti-inflammatory properties and is capable of decreasing the brain levels of TNF-α, IL-1β, IL-6, IL-4, IL-12, high mobility group box 1 (HMGB1), and NF-kB p65, but also increasing the level of anti-inflammatory cytokine IL-10 in the injured brain tissue [86,120,121] and significantly decreasing microglial activation after the TBI [121].

Resveratrol has been shown to substantially decrease nitric oxide (NO) and MDA, while increasing SOD and GSH protein levels when administered as a single injection shortly before [86] or immediately after the brain injury [122]. When given in the form of multiple injections after the TBI, resveratrol substantially increased GPx and 8-hydroxy-2′-deoxyguanosine (8-OhdG) protein levels in the injured tissue [123]. Moreover, when given orally in a low dose after the TBI, it has been shown to decrease ROS generation [124]. As was previously shown for several other (poly)phenols, resveratrol increased Nrf2 levels and, subsequently, HO1 levels in the injured brain [124].

It has been shown that resveratrol significantly decreased cortical contusion volume and preserved CA1 and CA3 hippocampal neurons from loss, maintaining the number of CA1 and CA3 hippocampal neurons at the level of non-injured animals [125,126]. A study by Shi and colleagues demonstrated that resveratrol, when given orally in a low dose, alleviates apoptosis and decreases the levels of cleaved caspase-3 in the injured tissue [124]. The study by Feng and colleagues demonstrated that resveratrol significantly reduced the levels of autophagy marker proteins, LC3II and Beclin1, in the hippocampus. Furthermore, the levels of TLR4 and its known downstream signaling molecules, NF-κB, and pro-inflammatory cytokines were also decreased after resveratrol treatment, suggesting that the neuroprotective effect of resveratrol is associated with the TLR4/NF-κB signaling pathway [120]. It has been shown that resveratrol pre-treatment inhibited ROS generation, the activation of inflammasome by decreasing NLRP3, and caspase1 levels as well as the IL-1β and IL-18 mRNA and protein levels while promoting SIRT1 activation [86]. 

Resveratrol administered either before or post-injury significantly reduced brain edema following the TBI [86,119]. Resveratrol has been shown to significantly enhance functional recovery after the TBI, as revealed by the improved post-injury mNSS, as well as the improved motoric score [120,125]. In addition, resveratrol also showed beneficial effects on spatial cognitive function after the brain injury as reflected in the significantly improved Morris Water maze performance of treated rats [120,125]. Moreover, animals treated with resveratrol displayed improved locomotor activity, reduced anxiety, and improved cortex/hippocampus-dependent memory after the TBI [126].

All the studies using resveratrol, except one [86], explored post-TBI treatment (Figure 4 and Appendix A). Like in the case of all (poly)phenolic compounds described in this review so far, resveratrol was very potent in reducing secondary brain injury and improving the functional outcome after the TBI regardless of the administration route and dose. However, as far as we know, no comparative study has been performed so far exploiting pre-, post-, and continuous treatment paradigms, to reveal which one gives the best results.

#### 3.1.8. Phenolic Acids

Phenolic acids (PA) represent a big subfamily of (poly)phenolic compounds. So far, several PA were tested in TBI settings, and all of them showed very strong effects on the processes of neuroinflammation, oxidative stress, and neurodegeneration, as well as on the functional outcome after the TBI (Figure 4 and Appendix A). One important aspect for this class of compounds is that besides their presence in the food matrix, some of these compounds are also gut microbiota catabolites of more complex polyphenols, such as, for example, 4′-hydroxy-3′-methoxycinnamic acid (ferulic acid), cinnamic acid, and 3′,4′-dihydroxycinnamic acid (caffeic acid). Therefore, the effects observed from this class may also mirror the effects of dietary parent (poly)phenols by oral administration and submitted to gut metabolism.

Gallic acid (3,4,5-trihydroxybenzoic acid) has been shown to significantly reduce TNF-α, IL-1β, and IL-6 protein levels in the brain following the TBI [79,80,81,121]. A study using 3,4-dihydroxybenzoic acid (protocatechuic acid) showed that this compound attenuated microglial activation in injured brain tissue and decreased the GSH depletion, contributing to a reduction in the secondary injury [127]. Remarkably, the use of the acute injection of ferulic acid showed a reduction in the neuronal apoptosis, increasing the levels of SOD and GSH after the TBI [128]. Both ferulic acid and protocatechuic acid have been shown to restore GSH and SOD levels while gallic and ferulic acid significantly decreased MDA levels in the brain tissue after the trauma [88,127,128]. A study by Nasution and colleagues showed that caffeic acid phenethyl ester (CAPE, a small chemical derivate of caffeic acid) treatment reduces oxidative damage as revealed by the reduction in the F2-IsoPs in the serum of injured animals [129]. Treatment with CAPE completely restored GPx, SOD, and MDA levels after the TBI [130]. 

When applied immediately after the TBI, protocatechuic acid was capable of substantially reducing neurodegeneration and dendritic loss in both the CA1 and CA3 regions of the hippocampus, as well as in the cortex [127]. Ferulic acid has also been shown to significantly alleviate neuronal cell loss in the CA1, CA2, CA3, and DG regions of the hippocampus as well as in the pre-frontal cortex of rats subjected to TBI [128]. Moreover, cinnamic acid has been shown to restore synaptic abnormalities by increasing synaptic spine density and synaptophysin protein levels [131]. Guo and colleagues suggested that cinnamic acid achieves these beneficial effects by suppressing the activity of HDAC2 through the promotion of the miR -455 -3p, a miRNA that regulates HDAC2 at the post-transcriptional level [131].

Treatment with cinnamic acid has been shown to significantly reduce brain edema [131], while treatment with a single injection of CAPE significantly reduced cortical contusion volume in rats [132]. A study by Nasution and colleagues showed that treatment with CAPE reduces AQP4 mRNA expression in the brain and AQP4 protein levels in the blood, suggesting this as a mechanism of action of CAPE in alleviating brain edema [133]. Post-injury CAPE administration also reduced TBI-associated vascular dysfunction by increased BBB integrity via the upregulation of occludin-5 levels [132]. Gallic acid treatment led to memory improvement [87,88,89,134]. Treatment with cinnamic acid alleviated spatial learning and memory impairments, as well as fear memory impairments [131]. However, CAPE treatment did not improve performance in either vestibulomotor/motor function or in learning and memory (Zhao et al., 2012). All the studies reviewed herein using phenolic acids in the TBI paradigm, except two [88,89], exploited the post-TBI paradigm. Based on our records, no comparative studies performing pre-TBI, post-TBI, and continuous administration of any phenolic acid have been performed so far. 

One major difference among other classes of (poly)phenols and phenolic acids should be considered when considering treatment with these compounds. Namely, phenolic acids are considered both parent compounds and metabolites at the same time. Since they occur as bioactive compounds in fruits, vegetables, and mushrooms, they can be classified as parent (poly)phenol compounds. However, they can also appear in the organism as gut metabolites of other, more complex parent (poly)phenols once these are ingested. Therefore, both oral application and intraperitoneal injection, in this specific case of phenolic acids, can have implications as a nutritional approach.

#### 3.1.9. Ellagitannins

In the TBI model, the subfamily of ellagitannins is represented only by one representative—ellagic acid—which has been shown to significantly reduce TNF-α, IL-1β, and IL-6 protein levels in the brain following the TBI [87,88,89,134]. A study by Mashhadizadeh and colleagues also showed that ellagic acid can reduce BBB permeability after the TBI [134]. Ellagic acid also managed to restore electrophysiological parameters and to improve the neurological score, as well as cognition and memory in both application regimens—as a pre- and post-treatment (Figure 4 and Appendix A) [134].

So far, only one TBI animal study using (poly)phenol-rich whole food has been performed. In this study, Sprague Dawley male rats were maintained on a diet supplemented with blueberry (BB, 5% *w*/*w*) for 2 weeks after fluid percussion injury [135]. This supplementation resulted in a significant increase in BDNF levels after the TBI and restoration of the BDNF-related plasticity marker—CREB and CaMKII—phosphorylation levels in the brain. By restoring the levels of 4-HNE, which is the end product of lipid peroxidation affecting plasma membrane integrity and neuronal survival and function, blueberry supplementation was proven to be able to counteract TBI-induced oxidative stress [135]. This study also showed that blueberry supplementation can improve functional outcomes after the TBI as shown by reversed learning and memory deficits after TBI.

### 3.2. (Poly)phenols Potential against TBI in Humans

There has only been one clinical trial performed so far exploring the protective effects of (poly)phenols in TBI patients. In 2021 [136], Zahedi and colleagues conducted a study where a daily dose of 500 mg of curcuminoids was administered to the TBI patients via enteral nutrition for 7 consecutive days. Curcumin supplementation has been shown to affect inflammation since the serum levels of IL-6, TNF-α, monocyte chemoattractant protein-1 (MCP-1), and C-reactive protein (CRP) were significantly reduced in these patients. Moreover, Acute Physiology and Chronic Health Evaluation (APACHEII) and Nutrition Risk in Critically Ill (NUTRIC) scores were significantly improved following curcuminoid consumption in comparison with the placebo. However, curcumin supplementation did not affect oxidative stress since no notable change was detected in the serum activities of GPx and SOD. These results are very promising and open a new avenue for future clinical trials exploring the effects of different (poly)phenols on the processes of secondary brain injury and overall outcomes after the TBI.

## 4. Gaps and Further Research Directions

(Poly)phenols are the phytochemicals that have shown to be one of the most promising bioactive compounds so far. The ability of (poly)phenols to interact with a variety of physiological processes, making them pleiotropic agents, suggests that including supplementation with (poly)phenols as part of the management of TBI might be a realistic approach to promote protective mechanisms in the injured brain. All data presented within this review, regardless of the species, strain, sex, and age of the animals, but also regardless of the TBI model and application route used, undoubtedly reveal the huge potential of (poly)phenols to successfully combat the secondary injury cascade of TBI at multiple levels (Figure 5). Utmost, the majority of studies showed significant effects on the outcome after the TBI in terms of improved neurological score, and motor and cognitive performance, when animals were treated with (poly)phenols. However, some issues must be addressed in the upcoming pre-clinical studies to provide more information and set good foundations for transferring the use of (poly)phenols in clinical trials in the future. 

Namely, almost all studies performed so far using (poly)phenols in the TBI setting used male animals. Although the results from the studies provide promising results in both sexes, we should be aware of sex differences in both pathological processes after the TBI, but also in response to bioactive compounds [137,138]. Therefore, more studies using both males and females are needed to be able to compare the efficiency of the treatment in TBI settings in each sex before making a conclusion based on the results mostly obtained in male animals. In addition, most of the studies researching the effects of (poly)phenols in the TBI model used the pharmacological approach, by giving a single (poly)phenol in very high doses by IP injection, in most of the cases applied after the TBI. To date, fewer studies have studied the oral administration of (poly)phenols in the TBI model, this administration route being closer to a nutritional approach. When (poly)phenols are applied orally, they are metabolized by gut microbiota, giving rise to numerous (poly)phenol metabolites. It has been shown that a single ingested dietary parent (poly)phenolic compound can give numerous (poly)phenol metabolites [139]. These metabolites are the ones which reach the circulation and organs [54,139]. On the other hand, when applied intraperitoneally, (poly)phenols skip the gut metabolism but are metabolized in the liver. Therefore, it should be taken into account that these two different application routes will lead to a different (poly)phenol metabolite composition that reaches the blood and subsequently organs, including the brain. However, although pre-clinical TBI studies showed undoubtedly positive effects of (poly)phenols in TBI settings, only one study performed metabolomic analysis to reveal the circulating (poly)phenols, as well as the ones reaching the brain, that could be responsible for the observed beneficial effects after (poly)phenol treatment [107]. Thus, the exact (poly)phenol metabolites and their mechanism of action, responsible for these effects, remain hugely unknown.

Another aspect of (poly)phenol treatments used in animal studies so far (both IP and oral administration studies) is the utilization of a single (poly)phenol compound. Namely, dietary (poly)phenols are ingested daily in the human diet, which are translated into circulating concentrations of (poly)phenols, mainly polyphenol-derived metabolites, and some reach the brain after passing through the BBB. Since different (poly)phenols can regulate different molecular pathways, using a cocktail of (poly)phenols, rather than a single compound, will give rise to the multitude of circulating (poly)phenol metabolites that could have better effects on the pathological processes of secondary brain injury, leading to a better outcome after the TBI.

Moreover, pre-clinical TBI studies, per se, have numerous constraints. Points of weakness in pre-clinical studies are represented by the variability in the TBI model implemented, in the (poly)phenols tested, in the timing, dosages, and routes of administration applied, but also in the variety of molecular and/or neurocognitive parameters evaluated. There are very few comparative animal studies that used different experimental paradigms in terms of the time of application (pre-, post-, or continuous treatment) of (poly)phenols in the TBI models. Those, however, point out that the timing of application can be a very important aspect since pre-treatment has been shown to be more effective than post-treatment [80], while the study comparing all three paradigms showed that continuous treatment with (poly)phenols is by far the best approach in terms of the observed results [80]. This points out that the preventive lifestyle, in humans, by using (poly)phenol-rich food daily, could be beneficial in the TBI scenario. However, only one clinical study using (poly)phenols in TBI patients has been performed so far, and although the results are promising, more studies are needed to give us enough data to be able to draw conclusions about the effects of (poly)phenol use in the human population after the TBI. One should keep in mind that besides the neuroinflammation that has been successfully tackled by (poly)phenol treatment in clinical settings [136] and numerous secondary injury processes efficiently regulated within pre-clinical studies, there are still numerous unanswered questions. Namely, nosocomial infections commonly occur in patients following TBI and are associated with an increased risk of mortality and poor neurological outcomes [140]. There is an extensive body of evidence for the potent anti-bacterial and anti-fungal activities of (poly)phenols, and some (poly)phenols show a synergistic effect when combined with antibiotics and anti-fungal drugs, suggesting a promising alternative for therapeutic strategies against antibiotic resistance [141]. However, there are still human TBI studies missing, to show whether treatment with (poly)phenols can reduce nosocomial infections in TBI patients, and hence, reduce mortality and improve the outcome after the TBI.

Moreover, increased intracranial pressure remains the main cause of TBI-related death among patients [142]. Although numerous pre-clinical studies reviewed herein undoubtedly showed that (poly)phenols can reduce brain edema in rats and mice following TBI, there are still human studies missing to confirm the same result in TBI patients.

Therefore, (i) standardizing TBI models, (ii) testing the efficacy of administration of a cocktail of (poly)phenols or (poly)phenol-rich food, rather than a single (poly)phenolic compound, (iii) performing additional interventional studies with TBI patients and assessing missing information concerning (poly)phenol effects on processes affecting outcomes after the TBI, and (iv) performing metabolomic analysis to reveal the metabolites that are the potential effectors in (poly)phenol-induced improvements in TBI outcomes, should be the future path in research. In addition, more studies comparing different paradigms in terms of the time of application (pre-, post-, or continuous treatment) would give valuable information about the best timing for (poly)phenol treatment. By addressing these important issues, future studies will set the foundation for (poly)phenol-based approaches to be applied in clinics, potentially improving the quality of life of millions of people suffering from TBI worldwide.

## Figures and Tables

**Figure 1 ijms-24-08908-f001:**
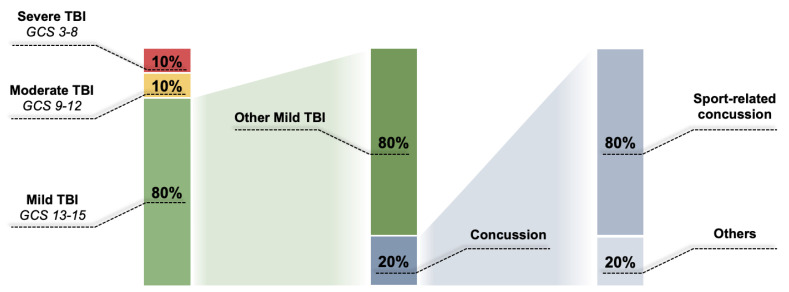
Traumatic brain injury (TBI) statistics worldwide. The vast majority of TBI cases are mild (Glasgow Coma Scale—GSC from 13 to 15). Of the total TBI, 80% are mild TBI. Of these, 20% are concussions; 80% of all concussions are sports-related concussions. The figure is created based on the epidemiological data previously published in [1].

**Figure 2 ijms-24-08908-f002:**
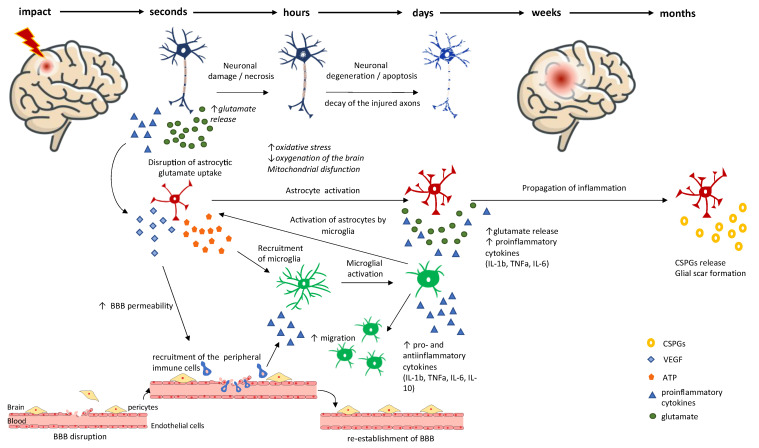
Primary and Secondary brain injury. Mechanical harm to the tissue at the moment of the impact induces processes of the primary injury—Blood–brain barrier (BBB) disruption, tissue deformation, shearing of neurons, axons, and glial cells and extensive glutamate release from neurons. These events start the cascade of secondary injury events. Due to BBB disruption, hemostasis, impaired oxygenation of the brain, brain edema, and further tissue damage occur. Decreased oxygenation induces oxidative stress that is characterized by mitochondrial dysfunction, rise of reactive oxygen species (ROS), and perturbed levels of main anti-oxidative enzymes (monoamine oxidase (MAO), superoxide dismutase (SOD), catalase (CAT), and glutathione peroxidase (GPx). In response to stimulation by the pro-inflammatory cytokines (predominantly interleukin (IL)-1β)) released by injured neurons, astrocytes generate and release vascular endothelial growth factor (VEGF) that increases BBB permeability and promotes leukocyte extravasation. These peripheral immune cells enter the brain tissue where they release pro-inflammatory cytokines and further propagate inflammation. Injured neurons release extensive amounts of glutamate which are normally uptaken by astrocytes. Injury disrupts astrocytic glutamate uptake, so excitotoxicity occurs. Damaged neurons die by necrotic cell death in the first hours after the injury which causes increased poly(ADP-ribose)-polymerase (PARP) levels. Parallelly, pro-inflammatory cytokines released from the injured neurons and ATP released by astrocytes recruit and activate microglial cells that migrate to the site of the injury and release extensive amounts of pro-inflammatory cytokines. Oxidative stress together with neuroinflammation leads to the second wave of neuronal death (apoptotic). This is characterized by an increase in caspase-3 cleavage and pro-apoptotic proteins (Bax, Bad) and a decrease in anti-apoptotic proteins (Bcl2, Bcl-xl). Microglial pro-inflammatory cytokines activate astrocytes that start releasing glutamate and pro-inflammatory cytokines which lead to further propagation of neuroinflammation. In the weeks and months after the injury, astrocytes form the dense glial scar around the injured area that represents physical, but also chemical obstacles for reinnervation of the injured region. Namely, astrocytes excrete chondroitin sulfate proteoglycans (CSPGs) making an inhibitory milieu for the neuritic regrowth.

**Figure 3 ijms-24-08908-f003:**
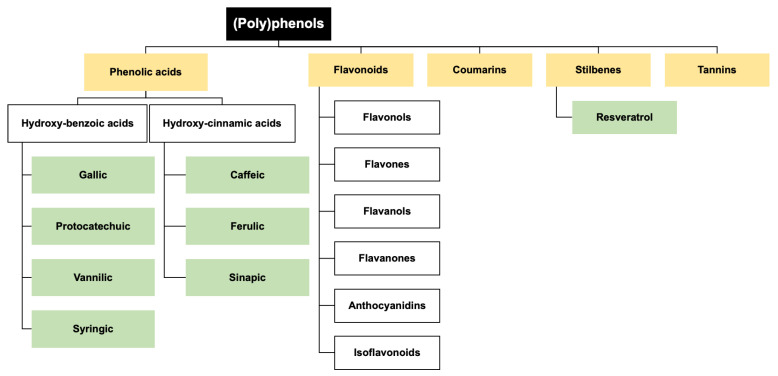
Main classes of (poly)phenols: Phenolic acids, Flavonoids, Coumarins, Stilbenes, and Tannins. In green are given some examples of compounds.

**Figure 4 ijms-24-08908-f004:**
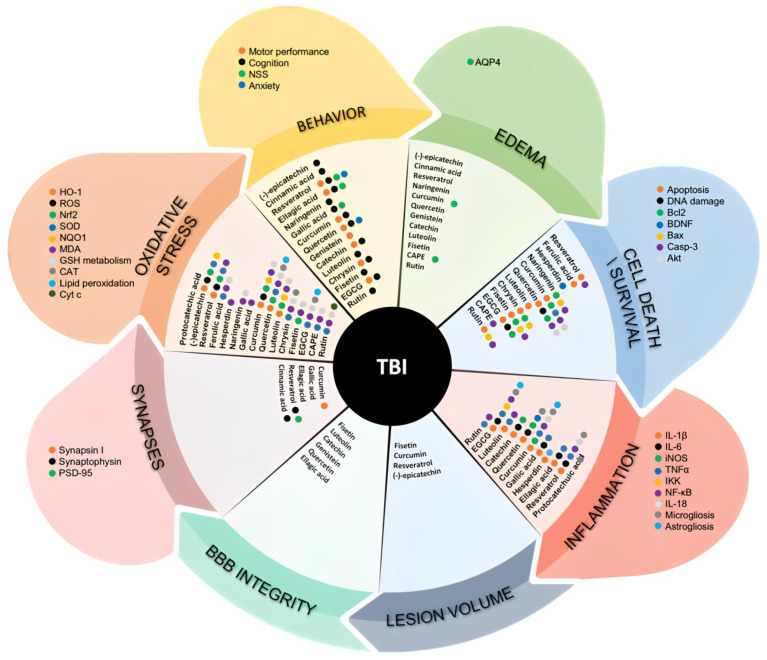
Evidence of (poly)phenol effects described on TBI models after oral administration. Main processes/targets impaired in TBI that were described to be ameliorated by (poly)phenols. TBI: traumatic brain injury; ROS: reactive oxygen species; TNF-α: tumor necrosis factor alpha; iNOS: inducible nitric oxide synthase; IL-1β: interleukin 1 beta; IL-6: interleukin 6; IL-18: interleukin 18; MDA: malondialdehyde; CAT: catalase; SOD: superoxide dismutase; Nrf2: nuclear factor-erythroid factor 2–related factor; HO-1: heme oxygenase 1; GSH: glutathione; NQO1: NAD(P)H:quinone oxidoreductase 1; NF-κB: nuclear factor kappa B; IKK: nuclear factor-κB (IκB) kinase; Cyt c: cytochrome c; PSD-95: postsynaptic density protein-95; iNOS: inducible nitric oxide synthase; NSS: Neurological Severity Score; AQP4: Aquaporin 4; BDNF: brain-derived neurotrophic factor; Casp-3: Caspase-3; Bcl2: B-cell lymphoma 2; Bax: Bcl2 associated X; Akt: protein kinase B.

**Figure 5 ijms-24-08908-f005:**
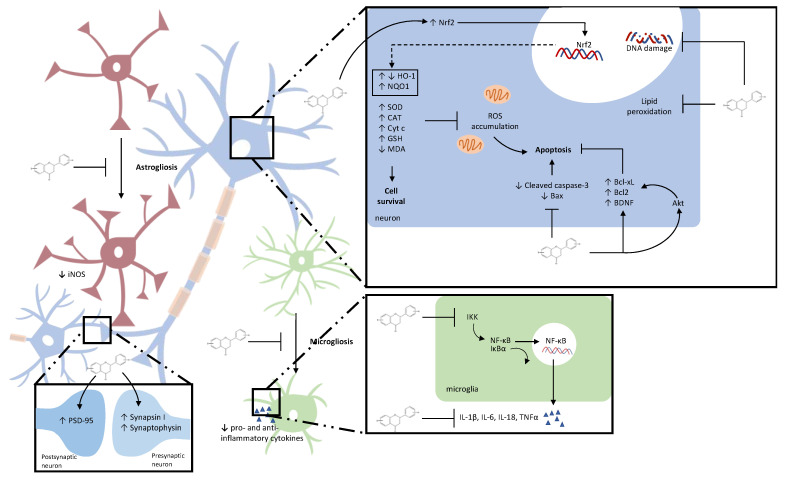
Main molecular effects of (poly)phenols on brain cells described in TBI models. This includes oxidative stress, inflammatory stress, and synaptic function. ROS: reactive oxygen species; TNF-α: tumor necrosis factor alpha; iNOS: inducible nitric oxide synthase; IL-1β: interleukin 1 beta; IL-6: interleukin 6; IL-18: interleukin 18; MDA: malondialdehyde; CAT: catalase; SOD1: superoxide dismutase 1; Nrf2: nuclear factor-erythroid factor 2–related factor; HO-1: heme oxygenase 1; GSH: glutathione; NQO1: NAD(P)H:quinone oxidoreductase 1; NF-κB: nuclear factor kappa B; IκB-α: nuclear factor of kappa light polypeptide gene enhancer in B-cells inhibitor; IKK: inhibitor of nuclear factor-κB (IκB) kinase; Cyt c: cytochrome c; PSD-95: postsynaptic density protein-95; BDNF: brain-derived neurotrophic factor; Bcl2: B-cell lymphoma 2; Bcl-xL: B-cell lymphoma-extra large; Akt: protein kinase B.

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
