# Peer review of "Dietary (Poly)phenols in Traumatic Brain Injury"

_ijms, 2023, doi:10.3390/ijms24108908_

Round 1
Reviewer 1 Report
In this manuscript, the authors reviewed the topic of dietary polyphenols in traumatic brain injury (TBI). The manuscript first provided an overview of TBI and its impact on the brain. Then it discussed the potential mechanisms by which polyphenols can exert their neuroprotective effects, including antioxidant and anti-inflammatory properties. The authors also reviewed several studies that have investigated the effects of specific polyphenols on TBI outcomes in animal models and human clinical trials. They concluded by highlighting some of the challenges associated with studying (poly)phenols in TBI research, including issues related to bioavailability and variability in (poly)phenol content across different foods. Overall, I agree with the authors that polyphenols may have beneficial effects on the outcomes of TBI, including the reduction of inflammation, oxidative stress, and neuronal damage. However, there are still some minor comments on this manuscript, and they are listed below for the author's reference.
1. In general terms, the authors seem to focus on the parts related to neuronal damage after TBI, including inflammatory responses and oxidative stress. However, clinically, there are also some less appreciatedaspects of treatment, including pain control, lowering brain pressure, and reducing infection. I wonder if polyphenols also have an impact on these smaller issues as the above?
2. All CNS drug use must first consider the efficiency of crossing the BBB. Although the authors mentioned this point in the later part, I would suggest that if they can further compare the differences in crossing the BBB efficiency of different types of polyphenols and provide evidence (it would be even better to organize them into a tab), I think that for some follow-up researchers in related fields will be helpful.
3. This article seems to seldom discuss the influence of polyphenols on ion channels (e.g. NMDA receptors). Since the progression of TBI seems to be related to these membrane receptors, I wonder what the authors think about this?
Author Response
Reviewer 1
In this manuscript, the authors reviewed the topic of dietary polyphenols in traumatic brain injury (TBI). The manuscript first provided an overview of TBI and its impact on the brain. Then it discussed the potential mechanisms by which polyphenols can exert their neuroprotective effects, including antioxidant and anti-inflammatory properties. The authors also reviewed several studies that have investigated the effects of specific polyphenols on TBI outcomes in animal models and human clinical trials. They concluded by highlighting some of the challenges associated with studying (poly)phenols in TBI research, including issues related to bioavailability and variability in (poly)phenol content across different foods. Overall, I agree with the authors that polyphenols may have beneficial effects on the outcomes of TBI, including the reduction of inflammation, oxidative stress, and neuronal damage. However, there are still some minor comments on this manuscript, and they are listed below for the author's reference.
1. In general terms, the authors seem to focus on the parts related to neuronal damage after TBI, including inflammatory responses and oxidative stress. However, clinically, there are also some less appreciated aspects of treatment, including pain control, lowering brain pressure, and reducing infection. I wonder if polyphenols also have an impact on these smaller issues as the above?
The reviewer raises very a valid point that we acknowledge. We addressed these issues by discussing them in subchapter 4. Gaps and Further Research Directions
2. All CNS drug use must first consider the efficiency of crossing the BBB. Although the authors mentioned this point in the later part, I would suggest that if they can further compare the differences in crossing the BBB efficiency of different types of polyphenols and provide evidence (it would be even better to organize them into a tab), I think that for some follow-up researchers in related fields will be helpful.
We kindly acknowledge the reviewer’s input on this very important topic of our research field, with which we fully agree. Nevertheless, besides the systemic effects that polyphenols and their metabolites arising from our diet may hold even if not BBB permeable, this point is indeed a very important aspect that we have not only studied already (10.1038/s41598-017-11512-6, 10.3390/nu11112678) but we have also been revising (10.3233/BPL-200099). Although this is not this review paper’s aim, we add further discussion on this topic to guide the reader for further details on other works and reviews focused on polyphenols and their metabolites BBB permeability, and for that reason, we will not include a table with the latest evidence here.
3. This article seems to seldom discuss the influence of polyphenols on ion channels (e.g. NMDA receptors). Since the progression of TBI seems to be related to these membrane receptors, I wonder what the authors think about this?
This is a valuable question and an interesting idea for further research. It is true that ion channels are particularly important players in the TBI aftermath since following TBI, the ionic homeostasis of the central nervous system becomes imbalanced. Excess Ca2+ influx into cells triggers molecular cascades, which result in detrimental effects. In recent years, Glutamate Receptor-Operated Ion Channels have emerged as a pivotal player in the pathophysiology of TBI, with their dysfunction resonating throughout the acute, subacute, and chronic phases of injury. With indiscriminate glutamate release in the acute phase, there is rising intracellular Ca++ that perpetuates neuronal injury (10.3171/jns.1998.89.4.0507.) and can lead to increased neuronal cell death and unfavorable outcome after TBI. On the other hand, TBI survivors exhibit long-lasting cognitive and affective sequelae that are associated with reduced quality of life and work productivity, as well as mental and emotional disturbances. While TBI-related disabilities often manifest physically and conspicuously, TBI has been linked with a “silent epidemic” of psychological disorders, including major depressive disorder (MDD). The emergence of MDD post-TBI may be rooted in widespread disturbances in the modulatory role of glutamate, such that glutamatergic signaling becomes excessive and deleterious to neuronal integrity, as reported in both clinical and preclinical studies (10.3389/fphar.2018.00579).
More than two decades the intensive research effort on the role of NMDA receptors (NMDAR) in traumatic brain injury (TBI) and cerebral ischemia (stroke) was led by these observations. Indeed, NMDAR antagonists were shown to improve post-injury recovery in animal models and subsequently, large-scale placebo-controlled clinical trials in TBI and stroke were performed with NMDAR antagonists. However, all these trials have demonstrated either no benefit or even deleterious effects (reviewed in 10.2174/18715273113126660196). This situation occurs often in TBI research when the drugs that are oriented towards one single process of secondary injury are transferred to clinics. Keeping in mind that (poly)phenols are pleiotropic agents, capable of affecting numerous processes of the secondary injury after TBI, one can speculate that they could also affect ion channels in this setting. However, to the best of our knowledge, the data about the role of (poly)phenols effects on ion channel functioning after TBI are still missing. Since there are data about the beneficial effects of (poly)phenols on ion channels in cardiovascular diseases (10.1007/s00210-022-02240-4; 10.1111/j.1755-5922.2010.00212.x), aging and age-related diseases (10.1016/B978-0-323-90581-7.00021-9) and protection of learning and memory (10.1097/WNR.0000000000001462), we strongly believe that (poly)phenols could exert the same beneficial effect in the TBI settings as well. Nevertheless, without existing data on the topic, it is hard to discuss this issue within the manuscript.
Reviewer 2 Report
The paper is interesting and well structured. The review is well constructed and the division into paragraphs makes for easy reading.
I only have two points to make.
1) in the introduction, paragraph 1.1, from line 55 to 78 is very scholastic and could well be shortened.
2) the literature cited is abundant but many papers published on the subject in this same journal have been ignored. For example:
Int J Mol Sci. 2023 Jan 30;24(3):2584. doi: 10.3390/ijms24032584.
Int J Mol Sci. 2022 Dec 22;24(1):199. doi: 10.3390/ijms24010199.
Int J Mol Sci. 2022 Oct 27;23(21):13000. doi: 10.3390/ijms232113000.
Int J Mol Sci. 2017 Dec 2;18(12):2600. doi: 10.3390/ijms18122600.
The paper needs moderate editing of English language.
Author Response
The paper is interesting and well structured. The review is well constructed and the division into paragraphs makes for easy reading.
We acknowledge very much the reviewer’s comment on our review paper.
I only have two points to make.
1) in the introduction, paragraph 1.1, from line 55 to 78 is very scholastic and could well be shortened.
We acknowledge the reviewer’s suggestion, which we agree with. In that regard, paragraph 1.1 has been revised and shortened.
2) the literature cited is abundant, but many papers published on the subject in this same journal have been ignored. For example:
Int J Mol Sci. 2023 Jan 30;24(3):2584. doi: 10.3390/ijms24032584.
Int J Mol Sci. 2022 Dec 22;24(1):199. doi: 10.3390/ijms24010199.
Int J Mol Sci. 2022 Oct 27;23(21):13000. doi: 10.3390/ijms232113000.
Int J Mol Sci. 2017 Dec 2;18(12):2600. doi: 10.3390/ijms18122600.
We appreciate the reviewer's suggestions. Suggested references in line with the scope of this review paper were included in the main text.
The paper needs moderate editing of English language.
The English language of the paper was carefully revised and edited, as recommended.
Reviewer 3 Report
This review by Carecho and colleagues describes the potential benefits of (poly)phenols in reducing the secondary cascade of injury happening after a TBI. The manuscript is divided in 2 parts, the first part describes the pathophysiology of TBI. The second part reviews studies which used (poly)phenols as a potential treatment for TBI (pre- or post-TBI). Both parts are well written and very complete, figures are very nice and clear.
The only thing I was wondering when reading the review was if these treatments showed differences in males vs females. It is mentioned in the discussion that treatments were effective regardless of sex, maybe a little more about this topic would be interesting to the reader, knowing that there are sex differences in TBI.
A few minor points:
-line 82: Should it be "20%"?
-line 116: Vascular Endothelial Growth Factor, not vasoactive
-line 219: Parkinson's
-Between line 259 and 266, the word "however" is repeated 3 times.
-line 276: Aquaporin
-line 297-298: tissue affected by TBI?
-line 392-395: It seems like there are extra words in the sentence.
-line 663: "in 2021"
Author Response
This review by Carecho and colleagues describes the potential benefits of (poly)phenols in reducing the secondary cascade of injury happening after a TBI. The manuscript is divided in 2 parts, the first part describes the pathophysiology of TBI. The second part reviews studies which used (poly)phenols as a potential treatment for TBI (pre- or post-TBI). Both parts are well written and very complete, figures are very nice and clear.
We kindly acknowledge the reviewer’s feedback on our manuscript.
The only thing I was wondering when reading the review was if these treatments showed differences in males vs females. It is mentioned in the discussion that treatments were effective regardless of sex, maybe a little more about this topic would be interesting to the reader, knowing that there are sex differences in TBI.
We acknowledge the reviewer’s suggestion. Very few studies (only 3 among all revised herein) done in female animals concerning the effects of (poly)phenols in the TBI model are showing the same beneficial effects in terms of the reduction of neuroinflammation, oxidative stress and improved behaviour (10.5137/1019-5149.JTN.17249-16.2; 10.1007/s10571-014-0070-9; 10.12659/MSM.909042). However, we do agree that this trend should be changed in future studies, to obtain more accurate and more applicable data.
We addressed this issue by discussing it in subchapter 4. Gaps and Further Research Directions.
A few minor points:
-line 82: Should it be "20%"?
-line 116: Vascular Endothelial Growth Factor, not vasoactive
-line 219: Parkinson's
-Between line 259 and 266, the word "however" is repeated 3 times.
-line 276: Aquaporin
-line 297-298: tissue affected by TBI?
-line 392-395: It seems like there are extra words in the sentence.
-line 663: "in 2021"
Minor points were amended in the manuscript, as suggested.
Round 2
Reviewer 2 Report
The paper is fine.